# The Case for the Target of Rapamycin Pathway as a Candidate Circadian Oscillator

**DOI:** 10.3390/ijms241713307

**Published:** 2023-08-27

**Authors:** Patricia Lakin-Thomas

**Affiliations:** Department of Biology, York University, Toronto, ON M3J 1P3, Canada; plakin@yorku.ca; Tel.: +1-416-736-2100 (ext. 33461)

**Keywords:** circadian, metabolism, Target of Rapamycin, yeast metabolic cycles, *Neurospora*

## Abstract

The molecular mechanisms that drive circadian (24 h) rhythmicity have been investigated for many decades, but we still do not have a complete picture of eukaryotic circadian systems. Although the transcription/translation feedback loop (TTFL) model has been the primary focus of research, there are many examples of circadian rhythms that persist when TTFLs are not functioning, and we lack any good candidates for the non-TTFL oscillators driving these rhythms. In this hypothesis-driven review, the author brings together several lines of evidence pointing towards the Target of Rapamycin (TOR) signalling pathway as a good candidate for a non-TTFL oscillator. TOR is a ubiquitous regulator of metabolism in eukaryotes and recent focus in circadian research on connections between metabolism and rhythms makes TOR an attractive candidate oscillator. In this paper, the evidence for a role for TOR in regulating rhythmicity is reviewed, and the advantages of TOR as a potential oscillator are discussed. Evidence for extensive feedback regulation of TOR provides potential mechanisms for a TOR-driven oscillator. Comparison with ultradian yeast metabolic cycles provides an example of a potential TOR-driven self-sustained oscillation. Unanswered questions and problems to be addressed by future research are discussed.

## 1. Introduction

The mechanisms of circadian (24 h) rhythmicity have been studied in a wide variety of organisms for over 60 years [1], and a consensus about the molecular mechanism that drives these rhythms, at least in eukaryotes, was formulated after the cloning of clock genes in the 1980s and the publication of the transcription/translation feedback loop (TTFL) model [2]. Decades of research have refined and broadened the TTFL concept, but there are still stubborn anomalies that challenge the TTFL model and require further development of our ideas about clock mechanisms. In this hypothesis-driven review, my goal is to pull together several lines of evidence around a potential non-TTFL oscillator candidate and evaluate its advantages and disadvantages as a component of circadian systems in eukaryotes. The Target of Rapamycin (TOR) signalling pathway, a ubiquitous nutrient-sensing pathway in eukaryotic cells, is an attractive candidate for a non-TTFL oscillator. It is both regulated by, and a regulator of, circadian rhythmicity, and provides a link between metabolic oscillations and the TTFL. Our laboratory is looking at the role of TOR in non-TTFL-based rhythms in *Neurospora*, and our results have strengthened the case for serious consideration of TOR as a bona fide circadian oscillator. An instructive comparison can be made between circadian clocks and ultradian metabolic rhythms in yeast and the role of TOR in generating these ultradian rhythms.

## 2. The TTFL Model of Circadian Oscillators in Eukaryotes

The “canonical” model for the TTFL mechanism of circadian oscillators in eukaryotes has been reviewed many times. Here, I will only cite a few key reviews for an overview of the genetic circuits, emphasizing the feedback loops at the core of the models to highlight comparisons with the feedback loops in the TOR pathway.

Although the *Drosophila* TTFL was the first to be established as a model for a circadian oscillator, here I will focus on the mammalian TTFL to provide context for the discussion of TOR effects (Figure 1a) [3]. In the core loop of the mammalian TTFL, heterodimers of the transcription factors CLOCK and BMAL1 bind to the promoters/enhancers of genes to activate their transcription. Targets of CLOCK–BMAL1 include the negative elements of the core TTFL, the three Per (period) genes, and the two Cry (cryptochrome) genes. The PER1, PER2, and PER3 proteins, along with CRY1 and CRY2, form complexes that inhibit the activity of CLOCK–BMAL1 in a negative feedback loop. Transcription of the Per and Cry genes then declines, protein levels decline, and the inhibition of CLOCK–BMAL1 is relieved to re-start the cycle. In a second feedback loop, nuclear receptor ROR activates BMAL1 expression, but is opposed by nuclear receptor REV-ERBα/β; transcription of REV-ERBα/β is, in turn, activated by CLOCK–BMAL1.

In the *Drosophila* TTFL [3], the basic logic is similar, but some proteins have changed functions. *Drosophila* Clk is the ortholog of mammalian CLOCK, but *Drosophila* Cyc (cycle) replaces BMAL1, so the positive transcriptional complex is Clk–Cyc. *Drosophila* Tim replaces CRY, so the negative complex is Per–Tim. In the second feedback loop, Clk–Cyc activates the transcription of two genes whose products, Pdp1e and Vri, regulate *clk* transcription. Cry functions as a photoreceptor in *Drosophila*, rather than as a transcription factor.

Representing the fungal kingdom is the filamentous fungus *Neurospora crassa* (Figure 1b) [4,5,6]. At the core of the TTFL is the *frq* (frequency) gene, whose transcription is activated by a heterodimer of two white-collar proteins, WC-1 and WC-2, forming the white collar complex (WCC). The FRQ protein fulfils the negative feedback function of the core loop by forming a complex with other proteins, including the FRQ-interacting RNA helicase (FRH), to inhibit the activity of the WCC by activating the phosphorylation of WC-1 and WC-2. The transcription of *frq* decreases, FRQ protein is phosphorylated and degraded, and the inhibition of WCC is relieved to restart the cycle. A second loop is formed through the positive feedback of FRQ on stabilization of WC-1 protein.

In the plant kingdom, research on the *Arabidopsis* circadian system has revealed multiple interlocked transcription loops [7]. Morning-phased genes CCA1 and LHY are repressors of evening-phased TOC1 and the ELF3/ELF4/LUX complex, while TOC1 represses CCA1 and LHY in a feedback loop. CCA1 and LHY activate PRR7 and PRR9, which inhibit the expression of CCA1 and LHY through a second negative feedback loop. The ELF3/ELF4/LUX complex represses PRR7, PRR9, and TOC1. This highly complex network is simplified in the alga *Ostreococcus tauri,* in which TOC1, LUX, and CCA1 are found but not the ELF genes, indicating that there may be only one feedback loop in this organism [8].

There is an abundance of evidence demonstrating the importance of TTFLs for circadian rhythmicity in many (but not all) eukaryotic cells. Mutations in TTFL components can affect the fundamental circadian properties of period, phase, and amplitude, indicating that these components are critical in regulating timing. TTFL disruptions, such as knockout mutants, can also disrupt many cellular rhythmic outputs, such as transcriptional and metabolic rhythms. It is clear from the descriptions of TTFLs across eukaryotic kingdoms that the identities of clock genes have not been evolutionarily conserved, although the logic of the basic transcriptional negative feedback loop is common. It has been repeatedly suggested [9,10,11,12,13] that this may point to the existence of an older, conserved oscillator mechanism that captured different transcriptional feedback loops in different lineages to stabilize the circadian system and/or provide additional output functions. A complete circadian system might require both non-TTFL and TTFL loops for optimal rhythmic functioning.

## 3. Rhythmicity in the Absence of the Canonical TTFL

### 3.1. Are TTFL Models Enough?

If the TTFL model is assumed to be the sole basis for the construction of cellular circadian oscillators, then the model fails in a number of cases where either TTFL components are non-functional or non-rhythmic (such as clock gene knockout or constitutive expression), or transcription in general is not functioning (in cells without nuclei, or when transcription is inhibited). There must be one or more non-TTFL-based oscillators driving rhythmicity in these systems, and the identities and mechanisms of these oscillators are unknown in eukaryotes. Milev et al. [13] provided an illuminating history of thinking about non-transcriptional oscillators from 1960 to the present, emphasizing the shift in attention towards TTFLs when clock genes were first identified through mutational screens in several model organisms. This concentration of research on TTFL mechanisms has delayed progress on investigating non-TTFL clock mechanisms, and the identification of the oscillator(s) that drive rhythmicity in the absence of functioning TTFLs remains a major unsolved problem in circadian biology.

The “second wave” of research on non-transcriptional oscillators has been focussed on “metabolic oscillators” as evolutionarily more ancient circadian clocks than TTFLs [12,13,14]. Note that in this review I am using “non-TTFL” oscillators, a somewhat broader term than “non-transcriptional oscillators”, to indicate an agnostic stance as to whether in some cells transcriptional feedback might be involved in putative oscillator mechanisms that do not require a functioning TTFL based on the well-characterized canonical “clock genes”.

### 3.2. Non-TTFL Rhythms

The prime example of a non-TTFL oscillator is the phosphorylation-based clock in cyanobacteria, in which the slow kinetics of kinase and phosphatase activities and protein conformational changes of the three Kai proteins create a 24 h cycle of gene expression, and this mechanism has been explored at the single molecule level [15]. In this review, we are confining the discussion to eukaryotic clocks, but the bacterial phosphorylation-based oscillator provides an example of how a non-TTFL oscillator can be constructed.

Non-TTFL circadian rhythms are referenced in a number of reviews, such as [12,13,16,17,18]. Examples of non-TTFL rhythmicity in eukaryotes include cells without nuclei, such as the single-celled alga *Acetabularia*, which can be experimentally enucleated and continues to rhythmically photosynthesize, and the human red blood cell, in which rhythms of PRX oxidation (see below) can be assayed. Cells without transcription or transcriptional rhythms include the single-celled alga *Ostreococcus tauri*, in which circadian timing information persists when transcription is stopped during prolonged darkness, and the dinoflagellate *Lingulodinium polyedrum* (formerly *Gonyaulax polyedra*), which has no detectable rhythms in transcript abundance yet displays well-studied rhythms in bioluminescence and photosynthesis [19]. As examples of non-functional clock genes in the animal lineage, mouse embryonic stem cells can continue rhythmic metabolism although the TTFL clock genes are not rhythmically transcribed, and behaviour rhythms can continue in *Drosophila* even when core clock genes are constitutively expressed. Recently, 24 h transcriptional rhythms were demonstrated in tissues from mice with a knockout of the core TTFL gene Bmal1 [20]. Knockout of mouse Cry genes does not abolish rhythms of PER protein in isolated fibroblasts [21]. In a recent example in plants, the proteome and phosphoproteome were assayed across a circadian time course in *Arabidopsis*, and it was found that a few proteins and phosphoproteins remained rhythmic in a strain with a disabled TTFL due to overexpression of the clock gene CCA1 [22].

The best-characterized extra-TTFL rhythm is the redox rhythm detected by assaying the oxidation state of peroxiredoxin (PRX) proteins, reviewed by [13,18]. This rhythm has been detected in a range of organisms in all domains of life [23]. PRX oxidation rhythms can be seen in organisms without functional TTFLs, including clock gene knockouts of *Drosophila* and *Neurospora*, and clock gene overexpression mutants of *Arabidopsis* and *Ostreococcus tauri* [23]. PRX rhythms are also seen when *O. tauri* is assayed under constant dark conditions where transcription is suppressed [24]. In isolated human red blood cells, which have no nuclei or organelles, PRX rhythms persist for several days under constant conditions [25]. PRX oxidation cycles are best thought of as a marker for circadian rhythms rather than potential oscillator components, since PRX gene deletions or PRX inhibitors do not abolish circadian rhythms in plants, bacteria [23], or mammalian cells [26].

In *Neurospora*, there have been a number of reports of rhythmicity in strains with knockouts in components of the TTFL, primarily in *frq* knockouts, but also in knockouts of the *wc* genes; these rhythms are known as FRQ-less rhythms and are presumed to be driven by one or more FRQ-less oscillators (FLOs). Examples from the year 1986 and onwards have been reviewed previously [6,16,27]. Our laboratory is interested in identifying the oscillator mechanisms that drive FRQ-less rhythmicity, using a genetics approach to identify putative components of the FLO. Using an unbiased mutagenesis screen, we found a mutation that abolishes FRQ-less rhythmicity under two different assay conditions [28]. The gene product (which we named VTA, vacuolar TOR-associated protein) was identified as a component of the TOR (Target of Rapamycin) pathway [27] and is homologous to a component of the EGO complex in yeast and Ragulator complex in mammals (see below). We have also studied a binding partner of VTA, one of the Rag proteins homologous to Gtr2 in yeast (see below) and have shown that a knockout of *gtr2* in *Neurospora* has a phenotype very similar to the *vta* knockout, abolishing FRQ-less rhythmicity [29]. These results have focussed our lab’s attention on the connections between the TOR pathway and circadian rhythmicity and raise the possibility that TOR is involved in the generation of FRQ-less oscillations in *Neurospora*.

## 4. The TOR (Target of Rapamycin) Pathway

### 4.1. TOR Complexes

The TOR pathway has been well-studied in mammals and yeast (*S. cerevisiae*) and is found ubiquitously in all eukaryotes. The following is a simplified overview (Figure 2), leaving out many of the TOR-associated proteins and regulatory mechanisms; for details, there are many excellent reviews of TOR signalling [30,31,32,33,34,35]. The TOR protein (mTOR in mammals) is a Ser/Thr protein kinase that is found in two different protein complexes: rapamycin-sensitive complex 1 (TORC1) and rapamycin-insensitive complex 2 (TORC2). Yeast, unlike most other organisms, has two TOR kinase genes: TOR1 and TOR2. TOR2 participates in both complexes, while TOR1 is only found in TORC1. In each complex, the TOR kinase is associated with several other proteins that regulate TOR activity, recruit substrates, and determine its subcellular localization. TOR complexes have emerged as central hubs for responding to environmental conditions and generating intracellular signals for growth and the control of metabolism. The roles of TOR in metabolic control can be considered at the cellular level, and also at the level of the organism when looking at tissue-specific functions [31], but the present discussion will focus on the functions of TOR at the cellular level and the general signal pathways that may be common in all eukaryotes.

### 4.2. Targets of TORC1 Output

Activation of TOR promotes cell growth and survival by activating anabolic processes and inhibiting catabolic processes. In both mammals and yeast, TORC1 phosphorylates substrates that, in turn, activate ribosome biogenesis and protein translation. In mammals, this includes the mTORC1 phosphorylation of S6K, which, in turn, phosphorylates ribosomal protein S6, increasing the expression of ribosome biogenesis genes, and the mTORC1 phosphorylation of 4E-BPs, which are translational repressors that are inactivated by mTORC1 phosphorylation, and thereby release translational initiation factor eIF4E to stimulate translation initiation [34]. In yeast, the S6K ortholog Sch9 similarly phosphorylates S6 protein [30]. Ribosome biogenesis in yeast is also activated by the TOR-mediated phosphorylation of the transcription factor Sfp1, which regulates ribosome biogenesis gene expression. Another major substrate of yeast TORC1 that activates protein synthesis is Tap42, which regulates phosphatases that dephosphorylate eIF2α to activate translation initiation [30].

Additional anabolic pathways activated by TOR in mammals include nucleotide and lipid synthesis [34]. Conversely, catabolic pathways can be inhibited by active TOR. In yeast, activated TOR represses the expression of stress-response genes [30]. The catabolism of intracellular material by autophagy is inhibited by activated TORC1 in yeast and mammals [30,34].

### 4.3. Regulators of TORC1

Signals that activate TOR complexes are stimuli for growth and survival. In mammals, insulin and growth factors are important TOR activators, acting through their plasma membrane receptors to inhibit TSC, an inhibitor of mTORC1 [34]. Yeast does not express TSC homologs, and nutrients are the only TORC1 activators. In both yeast and mammals, cellular energy status is signalled to TOR through the sensor AMPK, which responds to the cellular ADP/ATP and AMP/ATP ratios; glucose deprivation activates AMPK, which negatively regulates TORC1 [33].

Amino acids are major regulators of TOR activity, and amino acids from both extracellular sources and intracellular pools can be sensed by TORC1. In mammals, the mTORC1 complex translocates to the lysosomal membrane when activated by amino acids. The RAG GTPases, RagA and RagC, mediate this translocation and are required for the amino acid activation of mTOR. mTORC1 and the RAGs are anchored to the lysosomal membrane by the Ragulator protein complex. The pool of amino acids in the lysosomal lumen can activate mTORC1 through the Ragulator complex, and cytosolic amino acids can activate TOR through other sensors converging on the Rags [34]. In yeast, Gtr1 and Gtr2 are orthologs of the RAG GTPases in mammals and are similarly activated by amino acids, but through distinct sensors [33]. The Rags in yeast are anchored to the vacuolar membrane by the EGO complex, which is functionally similar to the Ragulator protein complex in mammals. TORC1 in yeast is constitutively localized on or near the vacuolar membrane, and amino acid pools in the vacuolar lumen are able to activate yeast TORC1 [33].

### 4.4. TORC2

The TORC2 complex seems to have a more evolutionarily divergent set of regulators and output functions than TORC1. TORC2 complexes are associated with the plasma membrane in mammals and yeast. In mammals, mTORC2 is primarily regulated by the insulin signalling pathway [34]. Stimuli that activate yeast TORC2 include stressors, such as high-temperature or hypotonic conditions [35]. The primary immediate target of TORC2 is the kinase Ypk1 in yeast [35] and its homologs in mammals and other organisms, and targets of Ypk1 are associated with the plasma membrane or adjacent ER membranes. This allows TORC2 to control aspects of membrane homeostasis, thereby regulating processes such as endocytosis, cytoskeletal remodelling, and cell migration [35]. TORC2 has also been reported to promote survival by inhibiting apoptosis. In mammals, mTORC2 contributes to inactivation of the mTORC1 inhibitor TSC [34]. The TORC2 complex does not form in plants, as they have lost the RICTOR protein required for complex formation [36].

### 4.5. The TOR Pathway in Plants

The components of the upstream and downstream TOR pathways have not been as well-characterized in plants as in mammals and yeast, but the core kinases and functions are conserved in the plant kingdom; see [36] for a comprehensive review. Glucose is an important upstream signal activating TOR in plants and appears to require the plant hormone auxin for its full activation. The sugar produced in the plant cell itself by photosynthesis is a major signal for TOR activation. Nutrients such as nitrogen and sulphur act upstream of TOR, and it is also activated by amino acids. The conserved TOR-S6K-S6 output pathway operates in plants to promote translation during morphogenesis and growth. Cell proliferation is activated by TOR through phosphorylation of transcription factors controlling cell cycle S-phase genes. Outputs also include supressing the catabolic process of autophagy. A compelling case can be made [36] for TOR as a central hub in plants for integrating a variety of external and internal signals to coordinate their growth and development in response to the variety of environmental conditions in which plants may find themselves.

## 5. TOR and the Circadian System

### 5.1. Criteria for Identifying an Oscillator

When we consider any candidate, such as the TOR pathway, as a potential component of the circadian system of a cell, there is no strict set of criteria that would allow us to declare it with certainty as “in” or “out” of the circadian oscillator mechanism. Roenneberg and Merrow [37] discussed the difficulty of drawing a distinct line between oscillator components and input or output pathway components that are rhythmically regulated and feed back to regulate the oscillator. If a process is both an input to an oscillator and a rhythmically regulated output, should it be defined as part of the oscillator mechanism?

There are varying roles that a component may play in a circadian system, with corresponding experimental predictions. The component may be part of an input pathway, such that a change in the component changes the phase of the oscillator and its output rhythms. The component may be a parameter, in which case its level would be constant (or at least not reproducibly rhythmic), but a change in its level or activity would affect the period and/or phase of the rhythmic clock outputs. The component may be an output of the system, in which case its level or activity would be rhythmic, but a perturbation of the component, such as blocking its activity, would have no effect on other clock outputs. If the component is an output of the TTFL-based oscillator, then alterations in the TTFL, such as clock gene mutations, would affect the rhythm of the component. If a component is a state variable of an oscillator, then its level or activity will be rhythmic; this rhythmicity will be required for rhythmicity of outputs, and any perturbation in the component will change the rhythmicity of all clock outputs. Because we are thinking about the circadian system of eukaryotes as likely consisting of several oscillators coupled to each other to create the complete system, we predict that perturbations of one oscillator will affect the others. It would be necessary to disable one of the oscillators, such as by mutation of an essential component, in order to reveal the properties of the coupled partner(s). Given this set of predictions, we can evaluate the evidence for the TOR pathway as a circadian oscillator.

Circadian rhythmicity is a property of individual cells and does not require interplay between tissues and organs in a complex organism to generate rhythmicity. Single-cell eukaryotic organisms are rhythmic, and single cells isolated from a tissue of an animal can be rhythmic. In order to evaluate TOR as a potential oscillator, we should focus on the evidence that comes from isolated cellular systems that are not influenced by signals from other cells, and that are studied under constant conditions in the absence of rhythmic environmental cues that could drive rhythmicity of cellular processes.

### 5.2. Is TOR Activity Rhythmic?

In looking for evidence of TOR rhythmicity to satisfy one of the criteria for an oscillator, we should only consider evidence from cellular systems that are not subjected to rhythmic hormonal signals from other tissues in a complex organism that could potentially drive rhythms in TOR activity through plasma membrane receptors. For example, feeding/fasting cycles in animals can be driven by the circadian oscillator in the brain and may, in turn, drive rhythmic TOR activity in peripheral tissues through changing the levels of glucose and hormones, but this does not provide us with an insight into the potential intrinsic rhythmicity of the TOR pathway in individual cells. TOR activity has been reported to be rhythmic in tissues removed from animals in light/dark cycles [38,39,40,41,42,43], or in constant darkness [44,45,46], but these TOR rhythms have not been shown to be cell-autonomous.

In *Arabidopsis*, TOR activity rhythms, observed by assaying phosphorylation of ribosomal protein S6, have been reported in whole seedlings under light/dark cycles and in constant light [22,47,48]. TOR activation in plants is closely linked to sucrose production by photosynthesis and transport of sucrose between source and sink tissues in the whole plant [49]. Coupling of circadian clocks between plant cells in the whole organism may occur through mobile molecules over both short and long distances [50], so it is unclear whether rhythms of TOR activity in whole plants could be considered cell-autonomous.

Reports of cell-autonomous rhythms in TOR activity are still sparse in the literature. Rhythmic mTORC1 activity has been reported in isolated synchronized mouse embryonic fibroblasts [43] and in a human breast cancer cell line synchronized by serum shock [51]. Cultures of dissociated chicken retinas that are highly enriched in one cell type, the cone photoreceptors, display rhythmic TOR activity, as assayed by the rhythmic phosphorylation of S6K and S6 [52].

Our laboratory has recently developed an assay for the phosphorylation state of endogenous S6 ribosomal protein in *Neurospora*, and we have detected a robust circadian oscillation of S6 phosphorylation in cultures growing on solid agar medium under constant darkness and constant temperature [53]. This implies rhythmic TORC1 activity in *Neurospora*. As described above, another important TORC1 substrate in yeast is TAP42, which activates phosphatases including Sit4, that in turn dephosphorylate and activate the translation initiation factor eIF2α. If the *Neurospora* TORC1 has similar targets as yeast, we would predict that the initiation of translation would be rhythmic in *Neurospora*. The Bell-Pedersen lab has extensively investigated rhythmic translation initiation in *Neurospora* by characterizing the regulation of the rhythmicity of phosphorylation of eIF2α [54,55]. Bell-Pedersen and colleagues have demonstrated roles for the rhythmic activities of the kinase CPC3 (homologous to mammalian GCN2) and phosphatase PPP-1 in rhythmically regulating the levels of eIF2α phosphorylation. Karki et al. [54] pointed out the possibility that other phosphatases, including the *Neurospora* homolog of Sit4, may also dephosphorylate eIF2α, although no evidence of this has been reported as of yet. Our peaks of S6-P at 28 and 52 h in darkness [53] coincide exactly with the phase of the minimum phosphorylation of eIF2α reported by Karki et al. [54]. This would be predicted if active TORC1 in *Neurospora* phosphorylated a TAP42 homolog to activate dephosphorylation of eIF2α, raising the possibility of rhythmic TOR activity as a contributor to rhythmic translation initiation in *Neurospora*.

### 5.3. Does the TTFL Influence TOR?

If the TOR pathway functions as an oscillator, we would expect to see mutual coupling between TOR and the TTFL within the same cell. A perturbation of the TTFL should affect the rhythm in TOR activity. Clock gene perturbations can affect TOR activity in tissues from clock gene knockout animals [45,46], but this may not indicate interconnections between the TTFLs and the TOR pathway in the same cell. Similarly, in *Arabidopsis*, a rhythm in TOR activity in whole seedlings in constant light was lost in a clockless strain overexpressing the clock gene CCA1 [48], but this may not indicate cell-autonomous control of TOR by the TTFL.

There are a few reports of cell-autonomous effects of the TTFL on TOR activity in animal cells. TOR activity was shown to be higher in isolated mouse lung fibroblasts from BMAL1 knockout mice [39], indicating that BMAL1 is a negative regulator of TOR. In primary mouse hepatocytes, HeLa cells, and mouse embryonic fibroblasts, mPER2 protein physically interacts with mTORC1 and suppresses its activity by recruiting TSC1 [43].

In *Neurospora*, we have assayed S6 phosphorylation in several clock-affecting mutant strains, and we have found that the activity rhythms of TOR match the periods and amplitudes of the clock output, as assayed by conidiation (spore-formation) rhythms on a solid agar medium [53]. In particular, the long-period *frq^7^* mutant increases the period and amplitude of both the conidiation rhythm and the S6 phosphorylation rhythm. The frq^7^ mutation directly affects the kinetics of negative feedback in the TTFL [56], and, therefore, this effect of *frq^7^* on S6 phosphorylation indicates that TOR rhythmicity is synchronized with the TTFL.

### 5.4. Does TOR Influence Circadian Timing in the Presence of the TTFL?

As reviewed by Cao [57], the circadian clock can be regulated by TOR through the following three mechanisms that have been observed in various mammalian systems: TOR may regulate the input pathway through which light entrains the phase and the period of the central clock in the brain; TOR may directly impact the clock mechanism in rhythmic cells; or TOR may influence the communication between rhythmic clock cells to affect their coupling into a network. TOR rhythmicity has been extensively investigated in mammalian tissues by Cao and associates, who have focussed on the SCN, the central clock in the brains of mammals that coordinates peripheral clocks throughout the body. Cao et al. have shown that TOR signalling regulates the light input pathway in the SCN via TOR control over protein synthesis [58], and TOR regulates synchrony of the coupled population of SCN neurons [59]. These effects of TOR can be localized to identified neuronal populations with cell-type-specific knockouts. When isolated mammalian cells are studied [60], knockdown of TOR in hepatocytes and adipocytes increases the period of PER2 expression. The activation and inhibition of TOR have opposite effects on the period and amplitude of PER2 expression in hepatocytes: activation decreases period and increases amplitude, while inhibition increases period and decreases amplitude [60]. In fibroblasts, hyperactive TOR leads to the increased expression of clock genes CRY1, BMAL1, and CLOCK [60].

In a different cell-autonomous mammalian system, U2OS cells, two different TOR inhibitors, rapamycin and Torin 1, can lengthen the period of PER2:LUC reporter gene expression in a dose-dependent manner [61].

In mice, body temperature becomes arrhythmic in most animals when TSC is lost from neurons [62], although a few rhythmic animals have a shortened period. The periods of the behaviour rhythms of heterozygous TSC+/− mice are also shortened, and this could be normalized with rapamycin treatment [62]. In mouse embryonic fibroblasts, TOR regulates the translation and turnover of BMAL1 protein; in mice, if the expression of BMAL1 is reduced, it reverses the effects of TSC loss on rhythmicity [62].

In *Drosophila* central brain clock neurons, the overexpression of TOR, its upstream regulator RHEB, or its downstream target S6K, will lengthen the period of the behaviour rhythm [63], in contrast to the shortened period of TSC+/− mice. The reduced expression of the TOR inhibitor TSC further lengthens the period in a background of S6K and RHEB overexpression, and when TSC was missing from central brain clock neurons, flies were mostly arrhythmic [63].

In *Arabidopsis*, the period of a circadian clock gene reporter is lengthened by depleting the plants of endogenous sugar, and TOR activity decreases at the same time; both effects are reversed by adding glucose [64]. Silencing TOR expression by RNAi also lengthens the period and blocks the reversal of period lengthening when glucose is added. A similar effect is seen with nicotinamide, which increases the period: the period lengthening is blocked by silencing TOR expression [64].

In *Neurospora*, the central TTFL protein FRQ is hyperphosphorylated in response to translation stress (treatment with the protein synthesis inhibitor cycloheximide, CHX) [65]. A previously described checkpoint kinase PRD-4 is responsible for this FRQ phosphorylation, and PRD-4 is, in turn, phosphorylated and activated by another kinase that is probably TOR [65]. This provides an explanation for the circadian phase-resetting effects of CHX in *Neurospora*: CHX activates TOR (see below in the “Protein synthesis feedback on TOR” section), which activates PRD-4, which phosphorylates FRQ. FRQ protein phosphorylation leads to its degradation, and this can potentially shift the clock phase.

In our lab, we have found that knockouts of two TOR pathway components in *Neurospora*, *vta* and *gtr2*, dampen the rhythm of conidiation and also damp out the FRQ protein rhythm [28,29], indicating that these gene products are required to maintain rhythmicity of FRQ protein. When taken together with the findings of Diernfellner et al. [65], this may indicate that TOR activity contributes to the rhythmicity of FRQ protein phosphorylation and stability.

### 5.5. Does TOR Participate in Circadian Timing in the Absence of the TTFL?

A crucial criterion for a non-TTFL oscillator is to demonstrate a role for the oscillator candidate in non-TTFL circadian rhythmicity. Evidence for TOR rhythmicity in the absence of a functional TTFL is very sparse. In one report, TOR activity, as assayed by S6 phosphorylation, was shown to be rhythmic in human breast cancer MCF-7 cells in culture. Although the cells were synchronized by serum shock, no rhythm in either mRNA or protein could be found for four essential clock genes BMAL1, PER2, REV-ERBα, or CRY2 [51].

*Neurospora* has provided evidence for a role for TOR pathway components in sustaining FRQ-less conidiation rhythms. The free-running rhythm of conidiation seen in FRQ-less strains under low-choline conditions can be abolished using knockouts of the two TOR pathway components *vta* and *gtr2*, and the entrained rhythm of conidiation under choline-sufficient conditions in FRQ-less strains is also severely impacted by these gene knockouts [28,29].

## 6. The TOR Pathway as a Self-Sustained Oscillator

### 6.1. The Relationship between TOR Rhythmicity and “Metabolic Oscillators”

As mentioned previously, current discussions around non-TTFL oscillators emphasize the role of “metabolic oscillators” in circadian timing. Stangherlin et al. [66] presented a view of circadian regulation in mammalian cells in which rhythmic TOR activity is an output of an unidentified “circadian timing mechanism”, with TOR activity driving rhythmic translation rates, which, in turn, drive many rhythmic cellular functions, including rhythmic metabolism. Meng et al. [36] proposed TOR as “a nexus to bridge the TTFL-driven and metabolism-associated clocks”. I would like to carry these models one step further and suggest that, at least in some cells, the TOR pathway may itself function as a “metabolic oscillator”.

### 6.2. Advantages of TOR as an Oscillator Candidate

The TOR pathway has several properties that make it an attractive candidate for a non-TTFL oscillator in eukaryotes. First, the TOR pathway is evolutionarily ancient in the eukaryotic lineage, and the core components were probably present in the last common eukaryotic ancestor [67], including both TORC1 and TORC2 components, as well as the amino acid-sensing RAG complex. There have been gene duplications (such as TOR1 and TOR2 in yeast) and losses (such as the TORC2 component RICTOR along with others in plants), but the evolutionary history of TOR signalling would place it early enough to be the basis for a common metabolism-centred oscillator in eukaryotes. Second, TOR could drive many rhythmic outputs, directly or indirectly. Stangherlin et al. [66] described the many consequences of the global increase in protein translation downstream of TOR activation and concluded that rhythmic TOR activity could account for many of the cell physiological rhythms. Third, interactions between TOR and the TTFL have been documented in several organisms (as described above), providing points of communication through which these two oscillators could become mutually coupled in a complete circadian system. Fourth, TOR is known to be regulated by negative feedback (described below), which is a common property underlying oscillator mechanisms.

## 7. Feedback Regulation of TOR Activity

### 7.1. Feedback Loops as the Basis for Oscillators

Although not strictly necessary for constructing an oscillator, a negative feedback loop with a time delay is probably at the core of many, if not most, biological oscillator mechanisms. TTFLs are all built around a genetic circuit in which a protein inhibits the transcription of its own gene. One property that makes the TOR pathway attractive as an oscillator is the documented existence of negative feedback loops regulating its activity (see Figure 2). Eltschinger and Loewith [32] have comprehensively reviewed many examples of feedback regulation of TOR activity, emphasizing “homeostasis” as the function of this regulation. Homeostasis, the maintenance of a relatively constant state, is incompatible with rhythmicity; clocks are constantly changing their state and are never static. However, the presence of a time delay in a feedback loop creates the minimal conditions required for generating rhythmicity rather than homeostasis.

The TOR complexes were initially described as responding to extracellular signals such as nutrition in the case of yeast, and both nutrition and growth factors in mammalian cells. In contrast, the concept of feedback control emphasizes the roles of intracellular signals in controlling TOR activity [32]. Feedback loops proposed for TOR regulation include the roles that TORC2 plays in lipid synthesis and homeostasis of plasma membrane tension [32], but the best-characterized feedback mechanism is based on the most obvious downstream target of TORC1 activation, which is the increase in protein synthesis via increases in ribosome biogenesis and translation.

### 7.2. Protein Synthesis Feedback on TOR

Feedback from protein synthesis to TOR is most clearly demonstrated by the activation of TORC1 by the protein synthesis inhibitor cycloheximide (CHX). The phosphorylation of ribosomal protein S6 in response to CHX treatment in mammals is a long-standing observation [68] and was cited in 1996 as evidence for the existence of feedback regulation of translation [69]. In yeast, the TOR substrate Sch9 (analogous to S6K1 in mammals) is rapidly phosphorylated in response to CHX treatment of cells [70]. In *Neurospora* [65], CHX treatment causes the phosphorylation of a hybrid S6 protein created as an artificial TORC1 activation reporter that is recognized by an antibody against mammalian phospho-S6. In *Neurospora*, CHX also leads to the dephosphorylation of the translation initiation factor eIF2α, a second indicator of TORC1 activation [65]. The signal for this feedback appears to be the level of intracellular, rather than extracellular, amino acids: CHX treatment of mammalian cells increases intracellular amino acids levels, and activation of mTOR signalling can occur in normally non-responsive amino-acid-deprived cells if these cells are treated with CHX [71]. The yeast EGO complex is localized on the vacuolar membrane, which is a major store of amino acids and nutrients, thus positioning EGO to respond to intracellular amino acid levels to activate TOR.

An additional source of feedback comes from ribosome biogenesis. The two TOR output pathways Sch9 and Sfp1 converge on the activation of ribosome biogenesis in yeast. Deletion of either of these genes increases TORC1 activity [72]. This may also be an effect of changes in amino acid pools as a result of decreased protein synthesis, or by another unidentified mechanism.

An alternate explanation of the effect of protein synthesis inhibitors on mTORC1 activity in mammalian cells is that inhibition of protein synthesis causes the loss of a short-lived protein inhibitor of TOR, REDD1 [73]. The REDD1 protein increases in response to extracellular stress and acts upstream of TSC2 to activate it, thereby inhibiting mTORC1 [74]. These two explanations for the effects of the inhibition of protein synthesis on TOR activity are not necessarily mutually exclusive and further illustrate the network of intracellular feedback mechanisms regulating TOR activity; both mechanisms may contribute to TOR activation, perhaps in different cell types.

### 7.3. Feedback from Autophagy to TOR

The increase in amino acids and TOR activity in cells treated with protein synthesis inhibitors highlights the complementary role of autophagy as a player in the metabolic feedback system, as discussed by Eltschinger and Loewith [32]. The increase in intracellular amino acids in response to CHX treatment [71] is assumed to derive from proteolysis that continues when protein synthesis is inhibited. The process of macroautophagy (or autophagy) degrades bulk material in the cytoplasm to recycle subunits, such as amino acids, for anabolic processes. Active TORC1 inhibits autophagy, and TOR inhibition by nutrient deprivation activates autophagy. However, in both mammalian cells [75] and yeast [76], prolonged nutrient starvation leads to the reactivation of TORC1 and the consequent repression of autophagy, thereby completing a feedback loop in which TOR is both upstream and downstream of autophagy.

## 8. Ultradian Metabolic Rhythms in Yeast

Research on the mechanisms behind ultradian (shorter than 24 h) rhythms of growth, cell division, and metabolism in yeast provides an instructive comparison to the search for non-TTFL mechanisms of circadian rhythmicity. Metabolic oscillations associated with the yeast cell cycle have been described by several groups. The yeast metabolic cycle (YMC) is an ultradian cycle characterized by alternating phases of low and high oxygen consumption coordinated with the stages of the cell division cycle and accompanied by transcription cycles. As noted by Papagiannakis et al. [77], metabolic cycles in yeast associated with the cell cycle were described as early as the 1960s, but this research depended on low-resolution studies on synchronized populations. Papagiannakis et al. [77] improved the methodology by looking at single cells in a microfluidic device and assaying NAD(P)H and ATP concentrations. Using a range of growth conditions and doubling times, these authors demonstrated persistent metabolic cycles even in the absence of cell division and concluded that the metabolic oscillations and the well-characterized cyclin/CDK cell cycle machinery form a system of coupled oscillators. Given the findings that the CDKs appeared late in the evolutionary history of eukaryotes, Papagiannakis et al. proposed that the metabolic oscillator appeared first and that the cyclin/CDK machinery evolved later to coordinate the cell cycle with metabolism.

Baumgartner et al. [78] used a system very similar to Papagiannakis et al., observing unsynchronized individual yeast cells in a microfluidic device, but using endogenous flavin fluorescence rather than NAD(P)H. These authors also demonstrated metabolic rhythms coupled to the cell cycle and oscillating independently under some conditions. They found that cycling continued in respiratory mutants, demonstrating that respiration is not necessary for metabolic cycling. Of particular interest in the context of the present discussion is the finding that the addition of the TOR inhibitor rapamycin lengthened the period of the metabolic cycle and decreased the regularity of the rhythm. These authors suggested that TOR could be involved in metabolic cycling.

Amponsah et al. [79] assayed peroxide levels across the YMC and found rhythms coordinated with the cell cycle. Perturbing the oxidation state of the cells with added peroxide or chemical oxidants or reductants could shift the phase of the cell cycle, and deleting two key peroxiredoxin genes abolished the rhythmicity of the cell cycle while maintaining the metabolic cycling, thus uncoupling the cell cycle from the metabolic cycle. As noted by O’Neill in the accompanying commentary [80], the persistence of metabolic cycling in peroxiredoxin-deleted strains highlights the role of redox oscillations as an output of rhythmic metabolism, not a cause of the oscillations. The mechanism behind the metabolic oscillations thus remains to be elucidated.

The most recent report [81] appears to be an example of a rhythm in TOR activity that is a self-sustained oscillation in cells under constant conditions. This work used yeast growing rapidly in rich (high-glucose) medium with a cell cycle of around 100–120 min. These authors were interested in looking for the mechanisms behind variations in growth rate during the cell cycle. Both TOR and protein kinase A (PKA) are known to regulate cell growth in yeast through the regulation of ribosome biogenesis and protein synthesis and are therefore good candidates for driving the oscillation in growth rate during the cell cycle. The authors found a peak of TOR and PKA activity in the G1 phase of the cell cycle. By mutating upstream regulators of TOR, they showed that a glutamine sensor, vacuolar protein Pib2, plays a role in generating the G1 peak of TOR activity. Mutations in upstream regulators of PKA that transduce signals of glycolytic flux also perturbed the oscillation of PKA activity during the cell cycle. These results suggest that upstream nutrient sensing is required for the generation of the oscillations in TOR and PKA activity. As the authors have suggested, since these cells are growing under constant conditions, there must be internally generated oscillations in metabolism to drive the downstream rhythms in TOR and PKA activity.

Stangherlin et al. [66], in comparing the yeast metabolic oscillations to circadian control of metabolism, placed TOR upstream of the switch to high respiration rates in the yeast cycle. Guerra et al. [81] suggested that metabolic oscillations drive the downstream oscillations of TOR and PKA activity. Given that TOR regulates energy-intensive processes of ribosome biogenesis and translation, it seems likely that an increase in TOR activity should lead to a depletion of energy charge and metabolite pools required for anabolic processes. These changes in metabolite levels should, in turn, affect TOR activity, as evidenced by the feedback mechanisms described above: inhibition of protein synthesis with CHX increases the levels of intracellular amino acids pools that then activate mTOR signalling [71]. The question then becomes: do metabolic oscillations drive TOR activity, or does TOR activity drive metabolic oscillations? Or could they all be components of a complex oscillating feedback network?

## 9. Problems and Unanswered Questions

### 9.1. Is TOR Activity Rhythmic in the Absence of a TTFL?

If TOR is to be considered as a candidate for a non-TTFL oscillator, an obvious requirement is to demonstrate rhythmicity of TOR activity under conditions where the TTFL is not functioning. We have only one report that is relevant, from human breast cancer MCF-7 cells in culture [51], showing that TOR can be rhythmic when canonical clock gene expression is not cycling. Assays of TOR activity in other non-TTFL systems should be a priority for testing this hypothesis.

### 9.2. Can the Kinetics of TOR Feedback Loops Account for 24 h Rhythmicity?

The self-sustained metabolic oscillations in yeast cells can show periods of 40 min to 5 h [78]. This is, of course, far from the 24 h period of circadian rhythms, and it is always dangerous to try to stretch the kinetics of ultradian rhythms to fit a circadian timescale. However, we do have examples of circadian metabolic oscillations that have been found under TTFL-less conditions. Among these, the prime example is the peroxiredoxin oxidation rhythm (described above), which is a marker for an unidentified metabolic/redox oscillation with 24 h kinetics. It should also be remembered that kinetics is an unsolved problem for the TTFL models as well, since it is still not possible to completely account for the 24 h kinetics of a circadian transcription/translation loop using known kinetic parameters for these processes [18].

### 9.3. What about Cells without TOR?

Not all cells that exhibit non-TTFL circadian rhythmicity have a functional TOR pathway. Peroxiredoxin rhythms have been documented in anucleate human red blood cells that contain no organelles and presumably no TOR pathway, and therefore the feedback loops involving TOR cannot account for rhythmicity in these cells. TOR may be only one of several or many non-TTFL oscillators.

### 9.4. Are TOR Effects on Rhythmicity “Merely Pleiotropic”?

It may seem trivial to suggest that TOR plays a role in supporting, or even generating, circadian rhythmicity, since TOR is central to metabolic control in cells, and everything ultimately depends on metabolism. TOR regulates protein synthesis, and proteins are required for the TTFL. Answering this objection would necessitate a demonstration that circadian rhythmicity of the cell requires rhythmic TOR activity, and not simply a permissive constant level of TOR function.

One hint that a permissive level of TOR activity is not enough for rhythmicity comes from our lab’s assays for S6 phosphorylation in *Neurospora*. In knockouts of two TOR upstream effectors, *vta* and *gtr2*, which participate in amino acid signalling to TOR in yeast, the S6 phosphorylation rhythm is gradually dampened in synchrony with the dampening of the conidiation rhythm that is measured as the physiologically relevant clock output [53]. The mean level of S6 phosphorylation (the mesor of the rhythm) is not significantly different between strains. These two knockouts also severely dampen the rhythm of FRQ protein levels [28,29], as previously mentioned. These results may indicate a requirement for rhythmic TOR activity to maintain a robust circadian system, and not simply a constant permissive level of TOR activity.

### 9.5. What Is the TTFL for?

If a metabolic oscillator, such as potentially the TOR pathway, can sustain rhythmicity and drive many outputs, what value does the TTFL add to the system? Many suggestions have been made, and it may be that TTFLs serve different functions in different cell types and organisms.

The output of the TTFL is rhythmic transcriptional control, and it has been suggested that this is the primary role of the TTFL in circadian systems. An evolutionarily ancient metabolic oscillator may have co-opted a transcriptional feedback loop to provide an additional mode of transcriptional control over cellular functions [12].

In contrast to the TTFL as an output mechanism, it may have also evolved as an input pathway [37]. The “zeitnehmer” concept proposes that the clock genes identified as TTFL components provide input from the environment, such as light, to set the phase of the core oscillator, which in turn contributes to regulating the levels of components in the TTFL “zeitnehmer” input pathway, making the TTFL both an input and an output of another oscillator.

Another function of the TTFL may be to provide the property of “compensation” to the circadian system [9]. “Compensation” is defined as the resistance of the oscillator to changes in period in the face of changes in the environment, such as different constant environmental temperatures or nutritional conditions. This is distinct from the acute effects of environmental changes on clock function, such as resetting the phase of the clock with an abrupt temperature change. For example, the *Neurospora* circadian system maintains approximately the same period in different constant temperatures, but mutations in the *frq* gene that change the period also change the property of temperature compensation [82]. A recent search for mutations affecting nutritional compensation in *Neurospora* [83] attributed the effects of the mutations to defects in the regulation of the expression of core TTFL clock genes. A prediction is that the metabolic or TOR-based oscillator on its own, in the absence of the TTFL, would be poorly compensated against changes in environmental variables such as temperature or nutrition. Indeed, FRQ-less conidiation rhythmicity seen in *frq* null strains exhibits poor temperature compensation [84,85,86,87] and poor nutritional compensation, such that the period of the FRQ-less rhythm is affected by the carbon source in the medium [84].

## 10. Conclusions

More than 30 years after the TTFL model was first proposed [2], we still do not have a satisfactory description of the circadian system of any organism at the molecular level, except for the phosphorylation clock of cyanobacteria [15]. Many anomalies remain unexplained, and the number of reports of non-TTFL rhythms continues to increase. There are currently no widely accepted candidates for a non-TTFL oscillator. Increasing interest in the interactions between circadian timing and metabolism puts a focus on regulators of metabolism, with the TOR pathway as a central player in metabolic control in all eukaryotes. It might repay our efforts to test the predictions of the hypothesis that, at least in some cell types, TOR and its feedback networks could be the elusive non-TTFL oscillator.

## Figures and Tables

**Figure 1 ijms-24-13307-f001:**
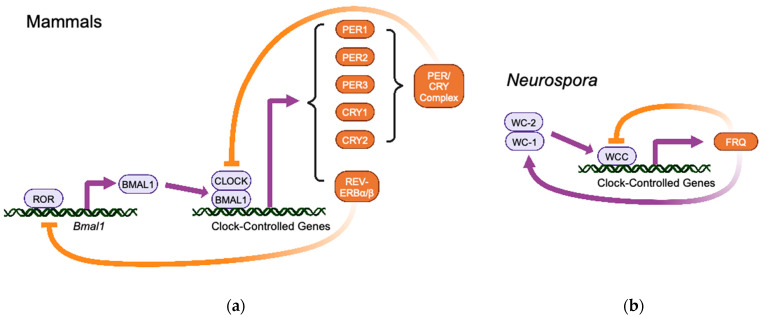
Simplified schemes of transcription/translation feedback loops at the core of molecular circadian oscillators. Not all components are included. Positive regulators are in purple; negative regulators are in orange. Thick curved arrows indicate feedback loops. (**a**) Mammalian TTFL. (**b**) *Neurospora crassa* TTFL. Created with BioRender.com.

**Figure 2 ijms-24-13307-f002:**
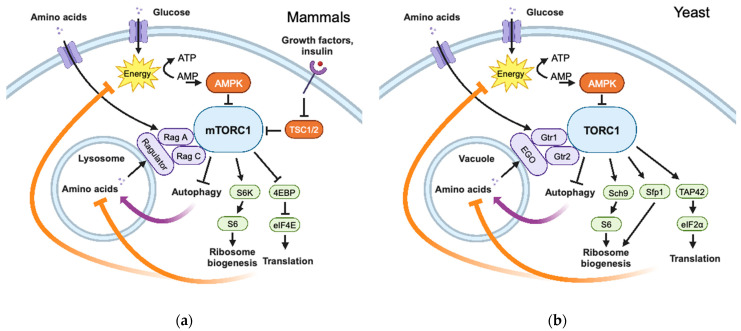
Simplified schemes of TOR pathways. Not all components are included. Plasma membranes and vacuolar/lysosomal membranes are indicated by thick double blue lines. Positive regulators are in purple; negative regulators are in orange; targets of TOR regulation are in green. Thick curved arrows indicate feedback loops. (**a**) Mammalian TOR pathway. (**b**) Yeast TOR pathway. Created with BioRender.com.

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
