# Peer review of "The Case for the Target of Rapamycin Pathway as a Candidate Circadian Oscillator"

_ijms, 2023, doi:10.3390/ijms241713307_

Round 1

Reviewer 1 Report

The manuscript by Lakin-Thomas is meant to be a review about TOR and circadian rhythmicity with an emphasis on the potential of the TOR system as an autonomous circadian oscillator. The author provides an overview of the relevant literature and puts the most prominent findings and concepts into context. They then move on to propose the novel hypothesis and conclude with a critical summary and open questions. The manuscript is well structured and uses clear language. The topic is reviewed comprehensively and, in some cases, puts old findings in an interesting new context. The novel concept is well explained and discussed in a broad context. All in all, it is a very enjoyable read and while I certainly do not agree with a number of conclusions (I am a much bigger fan of the TTFL than the author), thought-provoking articles are rare these days and this one should definitely be published. My main point is that in the current state this is not a review but a hypothesis paper. It should either be marked as such by the publisher or become more balanced, at least in the TTFL vs. non-TTFL part.

It is emphasized throughout the manuscript that the author finds the role of the TTFL in the generation of circadian rhythmicity to be less fundamental and non-TTFL mechanisms to have a bigger importance than generally presumed. This is a reasonable proposition for a hypothesis paper but for a review the manuscript is too biased on playing down the role of the TTFL. The best example is the statement in 94-99, which is far too strong in my opinion. The TTFL model does not “fail” in TTFL loss of function studies because non-TTFL-driven rhythmic output functions are in all cases only faint shadows of TTFL-driven rhythmic output functions. In other word, the manuscript completely lacks mentioning of the enormous consequences for cellular and organismal rhythmicity when the TTFL does not function. Numerous transcriptomic, proteomic, metabolomic, metabolic, physiological and behavioural studies exist in all important model organisms. Such studies together with a more quantitative view on the properties, robustness and regulatory power of TTFL vs. non-TTFL mechanisms would deserve quite some room in a balanced review article.

Minor points:

There is a long stretch of text (192—296) that is very sparsely referenced.

Much of the supporting evidence for the role of TOR in the circadian system stems from a single unreviewed publication (53, Akhtari thesis). This casts unnecessary doubts on the validity of the arguments. It might be worthwhile waiting with publication of this manuscript until a peer-reviewed paper has been published.

Reviewer 2 Report

This was an excellent review to read. The subject of non-transcriptional components in circadian oscillators has always fascinated me, and the author described the basis of clock architecture very clearly, before then detailing very clearly the different lines of evidence for oscillator function outside of TTFL. The writing style of this piece was to the point and refreshing. It was reminiscent of a scientific style that has in the main part been lost, the author is to be commended on this. My only content suggestion would have been to have explored TOR in plants, and the circadian-sugar signalling link, in a little more detail. However, it would change the scope of the article, and I do not recommend any changes, the fungi focus is the authors expertise. One thing that might be edited is in section 9.1 where the author writes "MCF-7 cells". As the author swaps between species frequently non-animal biologists might not know what species this cell line belongs to. In general the author has referenced the species consistently everywhere else. Thank you for the invitation to review this article, I enjoyed it immensely.
